# Liposomal Encapsulation of Citicoline for Ocular Drug Delivery

**DOI:** 10.3390/ijms242316864

**Published:** 2023-11-28

**Authors:** Claudia Bonechi, Fariba Fahmideh Mahdizadeh, Luigi Talarico, Simone Pepi, Gabriella Tamasi, Gemma Leone, Marco Consumi, Alessandro Donati, Agnese Magnani

**Affiliations:** 1Department of Biotechnology, Chemistry and Pharmacy, University of Siena, Via Aldo Moro 2, 53100 Siena, Italy; claudia.bonechi@unisi.it (C.B.); f.fahmidehmahdiza@student.unisi.it (F.F.M.); luigi.talarico@student.unisi.it (L.T.); simone.pepi@student.unisi.it (S.P.); gabriella.tamasi@unisi.it (G.T.); gemma.leone@unisi.it (G.L.); marco.consumi@unisi.it (M.C.); 2Centre for Colloid and Surface Science (CSGI), University of Florence, Via della Lastruccia 3, 50019 Sesto Fiorentino, Italy; 3National Interuniversity Consortium of Materials Science and Technology (INSTM), Via G. Giusti 9, 50121 Firenze, Italy

**Keywords:** citicoline, drug delivery, liposomes

## Abstract

Glaucoma represents a group of neurodegenerative diseases characterized by optic nerve damage and the slowly progressive death of retinal ganglion cells. Glaucoma is considered the second leading cause of irreversible blindness worldwide. Pharmaceutical treatment of glaucoma is critical because of the properties of the ocular barrier that limit the penetration of drugs, resulting in lower systemic bioavailability. This behavior causes the need of frequent drug administration, which leads to deposition of concentrated solutions on the eye, causing toxic effects and cellular damage to the eye. To overcome these drawbacks, novel drug-delivery systems, such as liposomes, can play an important role in improving the therapeutic efficacy of antiglaucomatous drugs. In this work, liposomes were synthesized to improve various aspects, such as ocular barrier penetration, bioavailability, sustained release of the drug, targeting of the tissue, and reduction in intraocular pressure. Citicoline (CDP-choline; cytidine 5′-diphosphocholine) is an important intermediate in the biosynthesis of cell membrane phospholipids, with neuroprotective and neuroenhancement properties, and it was used in the treatment on retinal function and neural conduction in the visual pathways of glaucoma patients. In this study, citicoline was loaded into the 1,2-dioleoyl-sn-glycerol-3-phosphocholine and cholesterol liposomal carrier to enhance its therapeutic effect. The citicoline encapsulation efficiency, drug release, and size analysis of the different liposome systems were investigated using dynamic light scattering, nuclear magnetic resonance, infrared spectroscopy, and ToF-SIMS experiments.

## 1. Introduction

Glaucoma, the most common optic neuropathy, is a heterogeneous set of progressively degenerative optic neuropathies and is characterized by the death or degeneration of retinal ganglion cells (RGC), which can progress to blindness [1]. It is a neurodegenerative disease involving ocular and visual brain structures [2]. The intraocular pressure (IOP) is considered the most important and the only modifiable risk factor for developing glaucoma because it can cause optic nerve damage in the retina. Although high IOP can cause the death of RGCs, the optic nerve damage may continue even after the treatment of IOP [3]. IOP is not a unique risk factor involved in RGC degeneration. Several events may occur as a sequence of pathological events to induce RGC degeneration leading to optic neuropathy, such as impaired microcirculation, ischemia/reperfusion damage, oxidative stress, neurotrophic growth factor deprivation, and mitochondrial destabilization [4].

Various directed or undirected strategies have been applied to protect RGCs and restore visual functions. RGC neuroprotection is the method that applies some agents such as citicoline to protect RGCs [1]. Citicoline (cytidine-5′-diphosphocholine) is an endogenous molecule essential to synthesizing membrane phospholipids and acetylcholine and increasing neurotransmitter levels in CNS. The neuroprotective properties of citicoline in glaucoma have been defined; in particular, it induces antiapoptotic effects, increases retinal dopamine levels, and counteracts thinning of the retinal nerve fiber layer [5]. It is an essential nutrient in the synthesis of membrane phospholipids in the central nervous system [6]. Ophthalmic drug delivery is a very important approach to optimizing the formulation of ophthalmic drugs and to avoiding the use of low-dose treatments with numerous side effects [7,8]. There are different topical ophthalmic drug forms, such as liquid, solid, and semisolid. However, other drug forms like multicompartment ophthalmic drug-delivery systems have some advantages [9].

Liposomes are one of the drug carriers in the multicompartment drug forms group [10,11,12]. Recently, liposomal ophthalmic formulations have been widely applied for drug-delivery applications, offering various advantages over the other ophthalmic delivery systems due to their biodegradability and nontoxicity. Also, their employment in ophthalmic drug forms enhances the bioavailability of loaded-molecule formulations against enzymes present on the surface of the corneal epithelium, while conventional ophthalmic formulations such as solutions lack bioavailability features [9,13]. Over the last decade, drug-loaded liposomes have been applied to treat various ocular disorders of both anterior and posterior segments. The lipids of vesicles surround an inner aqueous compartment. This feature makes them able to capsulate both hydrophilic and hydrophobic active molecules. The presence of cholesterol, a rigid and almost planar molecule, enhances the fluidity of the bilayer membrane and reduces the leakage of water-soluble molecules [14,15]. Water-soluble drugs like citicoline are capsulated into the inner aqueous part. The thin-film hydration method has been considered the most widely used technique [16]. The aim of the present research study was to encapsulate citicoline in a liposomal carrier with two different molar ratios, which is expected to improve the efficiency of drug delivery for possible ophthalmic applications. These vesicles are composed of 1,2-dioleoyl-sn-glycerol-3-phosphocholine (DOPC) and cholesterol (Chol) in two different molar ratios, (1:1) and (2:1), respectively [17,18]. The chemical characterization of the synthesized liposomes was evaluated in terms of particle size and surface charge, in vitro drug release, and physical stability.

The physicochemical properties of liposomes were studied to evaluate the behavior of citicoline in the presence of bilayers and to highlight its ability to deliver drugs. In particular, conformational properties were determined by nuclear magnetic resonance (NMR), attenuated total reflection Fourier transform infrared spectroscopy (ATR–FTIR), and time-of-flight secondary ion mass spectrometry (ToF-SIMS) experiments. Stability studies were also performed on all liposomal preparations to evaluate their storability as a pharmaceutical form.

These studies are important to evaluate the potential use of citicoline in liposomes for ophthalmic applications, with the aim of improving patient compliance.

## 2. Results and Discussion

### 2.1. Physicochemical Characterization: ζ-Potential and Liposome Size Distribution

The size distribution of liposomes plays an essential role in controlling the pharmacokinetics of liposomal drug formulations [19].

Particle size and zeta potential of liposomes are important parameters that should be considered during the preparation of liposomes as drug-delivery systems [20]. The results are shown in Table 1, and the corresponding distribution sizes for the DOPC:Chol liposomes are reported in Figure 1, as measured by dynamic light scattering. The DLS results showed that the mean particle size of plain liposomes, DOPC:Chol (1:1) and DOPC:Chol (2:1), were 118 ± 4 and 105 ± 3 nm, respectively. The binding of citicoline to liposomes as well as their nanoencapsulation increased the particle sizes, indicating an increase in the hydrodynamic diameter of liposomes due to the presence of a hydrophilic molecule in the system [21]. Moreover, the size of liposomes did not vary as a function of lipid composition (or molar ratio between lipids), and the different amount of cholesterol affected only the fluidity/permeability and the organization (phase and domain formation) of phospholipid bilayers [22].

In addition, the size of liposomes is the most important factor for targeting. Larger liposomes are mostly taken up by phagocytes, whereas small liposomes (<200 nm) can easily penetrate tumor tissue due to the effect of increased permeability and retention (EPR) [23].

ζ-potential is a key parameter that is widely used to predict the stability of suspensions. The ζ-potential of the plain liposomes was near neutrality. As the table illustrates, they showed a slight negative zeta-potential charge that can be due to preferential absorption of H_2_O ions from the water environment or to the outward exposure of phosphate groups. These values suggest that the vesicles are not more storable in solution: the aggregation or flocculation processes probably occur in solution. The greater the absolute value of the zeta potential is, the more stable the suspension formed is [24]. DOPC:Chol, Cit (1:1), 1 and DOPC: Chol, Cit (2:1), 1 exhibited a zeta potential close to zero, −2.2, and −1.8 mV, respectively.

The polydispersity index (PDI) is a measure of the width of unimodal size distributions [25]. In fact, small polydispersity indices of the prepared liposomes indicate good dispersion homogeneity (<0.2), suggesting that all liposomes are relatively monodisperse [26]. All liposome samples can be considered sufficiently monodisperse.

The DLS experiments show that the citicoline insertion in the zwitterionic liposome does not modify the physicochemical properties of plain vesicles.

### 2.2. Determination of Encapsulation Efficiency

Encapsulation efficiency (EE) is a critical parameter in measuring the quality of liposomes, which refers to the ratio of drugs entrapped in the liposomes to the total drugs added. In addition, the key to assessing encapsulation efficiency is to isolate the free drugs after the structure of the liposomes has been destroyed. The calculated encapsulation efficiency of each formulation is reported in Table 2. In this study, the free molecules were separated using a dialysis membrane (for 12 h) and were detected with a UV spectrophotometer at 272 nm (at 298.15 K). The encapsulation efficiency (EE%) of both formulations was calculated by Equation (1); see Section 3.

The EE% for the two liposome formulations DOPC:Chol, Cit (1:1), 1 and DOPC:Chol, Cit (2:1), 1 is 40 ± 11% and 47 ± 11%, respectively. The zwitterionic liposomes are efficient systems for the encapsulation of a hydrophilic drug such as citicoline, and the experimental synthesis procedure was correct to maximize this parameter. The higher cholesterol concentration in the liposome formulation also affects the EE% value: increasing the cholesterol concentration increases the release process of citicoline but decreases the encapsulation efficiency of the drug in the liposome system. Therefore, the greatest fluidity of the bilayer decreases the ability to encapsulate citicoline.

### 2.3. Release of Citicoline versus Time in the Simulated Ophthalmic Fluid

Figure 2 shows the free citicoline at 272 nm detected by a spectrophotometer that achieved its complete release within 6 h after separation by dialysis procedure using a membrane with a cutoff of 14 kDa in a simulated ophthalmic fluid for the two liposome formulations (1:1), 1 and (2:1), 1.

The experimental results show that the release of citicoline is 41% and 49% for DOPC:Chol, Cit (1:1), 1 and DOPC:Chol, Cit (2:1), 1, respectively, after 1 h of contact with simulated ophthalmic fluid. After 6 h, the concentration of citicoline released is 93% and 79% for DOPC:Chol, Cit (1:1), 1 and DOPC:Chol, Cit (2:1), 1 liposomes, respectively.

The formulation DOPC:Chol, Cit (1:1), 1 reaches a high value of release faster than the formulation DOPC:Chol, Cit (2:1), 1. Cholesterol concentration affects the release process of citicoline in physiological fluid. In particular, a higher cholesterol concentration accelerates the release of the drug. This behavior is directly influenced by the increase in fluidity of the lipid bilayer, which increases with cholesterol concentration [27].

In the regular shapes of liposomes, cholesterol determines a variation in membrane deformability and this behavior influences the release of citicoline.

In Figure 2, the fitting of kinetic data was also reported for the two different liposome formulations. A pseudo-first-order kinetic model well describes the mechanism of release of drug in both of the liposome formulations. This fitting is used to describe the release processes for the drug-delivery systems loaded with water-soluble compounds or to explain a controlled, predictable, and gradual release of the drug [28]. The experimental data were fit by the integrated equation y = A(1 − exp(−kt)), where y is the amount of drug released, k is the first-order rate constant, and t is the time. The fitting coefficients are A = 92.34 and 91.95 for DOPC:Chol, Cit (1:1), 1 and DOPC:Chol, Cit (2:1), 1, respectively, obtaining R^2^ factors of 0.998 and 0.999, respectively. The drug-release pattern followed a pseudo-first-order kinetics and increased with exposure time to a maximal release.

### 2.4. NMR Experiments

Solution state ^1^H NMR spectroscopy was used to elucidate the ability of citicoline interacting with liposome systems and to study the structural change due to its inclusion in the bilayer. In particular, ^1^H-NMR studies were carried out to confirm the presence of encapsulated drug molecules after purification. In Figure 3, the proton spectrum with assignment of citicoline 10^−2^ M in D_2_O solution is reported. Table 3 reports the proton chemical shift values with a multiplicity of signals. Proton assignment was determined by 2D experiments and for comparison with the literature data. These experimental data are important to verify the presence of citicoline in the liposome and to identify the chemical position in the bilayer.

Figure 4 show the proton spectrum of the empty DOPC:Chol (1:1) and DOPC:Chol (2:1) liposomes, respectively. The DOPC proton assignment, obtained using 2D dimensional NMR experiments, is reported in Table 4. The first experimental evidence is the absence of any difference, both chemical shift values and peaks intensity, for two different liposome formulations. The presence of different concentrations of cholesterol modified the fluidity of bilayers without affecting the NMR spectra. Large NMR proton signals confirmed the presence of vesicles in solution, and the unresolved peaks are a typical characteristic of aggregates [29]. Amphiphilic molecules that form aggregates exhibit slow molecular motions and incompletely averaged dipolar interactions within the aggregate, leading to drastic line broadening of NMR spectra.

Figure 5 shows the comparison between loaded and plain liposomes DOPC:Chol (2:1) and the spectrum of pure citicoline in solution. This approach allows us to highlight the presence of the active compound in the liposomes. In the spectrum of liposomes loaded with the drug, low-field proton signals attributable to citicoline protons (Ha, Hb, Hc) can be seen after dialysis. In particular, the aromatic signals, Ha and Hb, indicate the insertion of citicoline into the hydrophilic head of the phospholipid bilayer and confirm the successful encapsulation of citicoline in the vesicles. The broadening of the NMR signals of citicoline (Figure 5) directly indicates the correct incorporation of the drug into the liposome. This behavior is due to the slow reorientation motion that citicoline adopts once it is incorporated into the phospholipid bilayers. These NMR data are in agreement with the experimental values of the encapsulation efficiency of liposomes loaded with citicoline.

The NMR study carried out on the DOPC:Chol, Cit (1:1), 1 liposome shows an identical behavior, suggesting that different concentrations of cholesterol do not modify the insertion properties of the drug in the liposomes.

### 2.5. ATR-FTIR Studies

Pure citicoline, plain liposomes, and citicoline-loaded liposomes were subjected to ATR-FTIR studies between 4000 cm^−1^ and 750 cm^−1^ to verify the association between citicoline water-soluble compounds and phospholipids by analyzing the frequency of different vibrational modes. The loaded samples were subjected to purification prior to the experiment to remove the interference of water and free active molecules. Figure 6 shows the most important vibrations of the functional chemical groups of phosphatidylcholine highlighted in the FTIR analysis. Figure 7 and Figure 8 show the spectra of a DOPC: Chol before and after loading citicoline with two different molar ratios of DOPC and cholesterol, 1:1 and 2:1, respectively. Magnification of the spectra of the 1700–750 cm^−1^ region is shown in Figure 7 and Figure 8, presenting the most relevant bands [30].

Main absorption bands together with their assignment are listed in Table 5 [31].

The presence of citicoline in the purified liposomal formulation is confirmed by characteristic bands attributed to the active compound in the spectra of drug-loaded liposomes as the NH_2_ wagging motion at 787 cm^–1^, the appearance of a shoulder at 1117 cm^–1^, and as a component of the corresponding adsorption band of citicoline due to the stretching of C-O ether bond.

### 2.6. ToF-SIMS Measurements

For pure citicoline, plain ToF-SIMS analysis was performed to support NMR and IR data. Figure 9 and Figure 10 show, respectively, the main informative region (100–400 *m*/*z*) for the positive ion spectra obtained from a pure citicoline solution and DOPC/Chol, Cit liposomes. Due to the large similarity of functional groups between phospholipids and citicoline, only some observed fragments could be used as confirmation of the presence of citicoline in the described formulation. From the reference citicoline spectrum, characteristic fragments of the compound are identified as *m*/*z* 112, (cytosine+2H)^+^, *m*/*z* 227, (C_8_H_12_N_3_O_4_^+^), and the *m*/*z* 104 ion, corresponding to the choline moiety (Figure 9b). This last ion, together with *m*/*z* 353 (C_13_H_11_N_2_O_6_P_2_^+^) and *m*/*z* 381 (C_9_H_10_N_4_O_11_P^+^), could also be generated from phospholipids (Figure 10b). In the spectra of DOPC/Chol, Cit liposomes (Figure 10a), the presence of citicoline in the liposomal preparation could be confirmed only by the presence of an ion at *m*/*z* 112, as the other fragments are reconducible to both DOPC and cholesterol.

### 2.7. Physical Stability of Liposomes Loaded with Citicoline versus Time

The liposome is a system of thermodynamic instability; its physical and chemical properties may change during storage. The vesicles may gather and fuse, leading to the leakage of the drug and an increase in the particle size over the storage period.

In the present study, the mean size and PDI value of plain and loaded liposomes stored at 277 K in PBS solution for 3 months were evaluated to understand their stability versus time. As shown in Figure 11a,b, none of the liposomal formulations indicated substantial changes in hydrodynamic size and PDI within 3 months, indicating that the liposomes had enough stability. However, the PDI value of all four liposomes slightly increased over 3 months of storage.

Figure 11a reports the hydrodynamic size for all plain and loaded liposomes recorded versus time (0, 30, and 90 days). None of the liposomal formulations showed significant changes in hydrodynamic size within 3 months, suggesting that the liposomes were sufficiently stable and that the aggregation/flocculation processes did not modify the physical–chemical properties. Similar evidence can be obtained by analyzing the PDI parameters versus time. As reported in Figure 11b, the PDI value remained stable around 0.20 for empty liposomes and increased up to 0.4 for the vesicles containing citicoline. This suggests that empty liposomes were stable over time and relatively monodisperse, while citicoline liposomes showed a PDI of 0.30–0.35, which can be considered to be acceptable and indicates a homogenous population of phospholipid vesicles.

Moreover, the different cholesterol content in the two liposomal formulations does not affect the stability of the vesicles in PBS solution.

## 3. Materials and Methods

### 3.1. Plain Liposome Preparation

Liposomes were prepared using the thin-film hydration method. Liposomal formulations consisted of 1,2-dioleoyl-sn-glycerol-3-phosphocholine (DOPC) and cholesterol (Chol) at two different molar ratios: 1:1 and 2:1. DOPC was purchased from Avanti Polar Lipids (Alabaster, AL, USA). Cholesterol and solvents were purchased from Sigma Aldrich (Saint Louis, MO, USA).

Liposomes were prepared in a round-bottom vial by mixing the appropriate amounts of stock solutions in chloroform. A dry lipid film (without active molecules) was obtained by evaporating the solvent slowly under nitrogen flow and then completely removing under vacuum overnight. The lipid films were hydrated with PBS (D_2_O, for NMR experiments). To obtain uniform liposomes with reduced or no lamellarity, all resulting vesicles underwent two subsequent steps: (i) homogenization by ten freeze–thaw cycles, freezing in liquid nitrogen and warming in a water bath at 50 °C; (ii) extrusion through 100 nm polycarbonate membranes (27 passages) with a LiposoFast apparatus (Avastin, Ottawa, CA, USA). The final total lipid concentration was 10^−2^ M for all samples.

### 3.2. Drug-Loaded Liposome Preparation

Drug-loaded liposomes were prepared as described above. The hydrophilic drug (citicoline, 5.4 mg/mL in PBS) was added to the aqueous phase during the hydration processes of the lipid film, and the liposomes were obtained using the same two processes (freeze–thaw and extrusion). In Table 1, the phospholipids’ molar ratio is reported.

Purification was performed by dialysis procedure using a cellulose tubular membrane with a molecular weight cutoff of 14 KDa (Sigma-Aldrich Chemie GmbH PO, Taufkirchen, Germany).

### 3.3. Determination of Encapsulation Efficiency

The encapsulation efficiency of liposomes (EE%) was determined by measuring the amount of encapsulated molecules with a UV–visible spectrophotometer after liposome destruction (Lambda 25 UV/VIS Spectrophotometer, a product of PerkinElmer Inc. spectrophotometer Waltham, Massachusetts, USA, in a 1 cm quartz cuvette). The absorbance was measured at 272 nm, corresponding to the wavelength of maximum absorption (λ_max_) of citicoline. This λ_max_ was obtained by scanning the absorbance of the drug between 200 and 800 nm. A calibration curve constructed for Cit in PBS (range: 40–130 μg/mL, with the linear equation y = 0.0152x − 0.0291 and R^2^ = 0.9997) was used to quantify C_it_ in the liposomal preparation to calculate the percentage EE and loading. This experiment studies the effect of cholesterol concentration on EE% of capsulated drug molecules. Encapsulation efficiency was calculated by using the following Equation (1) [32]:(1)$$EE\%=\frac{C_{f}}{C_{i}}\times 100$$

C_i_ is the concentration of the initial citicoline added, and C_f_ is the concentration of detected free active molecules.

### 3.4. Dynamic Light Scattering (DLS)

The physicochemical properties of liposomes were evaluated using the dynamic light scattering (DLS) technique for size distribution and ζ-potential. The analysis was determined at pH 7.4 (PBS buffer) and 298.15 K using dynamic light scattering (DLS) and a Zetasizer Nano ZS90 instrument (Malvern, UK) supplied with a He/Ne laser at a 633 nm wavelength.

All liposome samples were diluted with PBS buffer solution before the DLS experiments. The samples were equilibrated at 298.15 K for 5 min before measurement. All solution samples were passed through a PES syringe filter (pore size = 0.45 μm) prior to DLS experiments, and all measurements were performed in triplicate.

### 3.5. Nuclear Magnetic Resonance Experiments (NMR)

NMR experiments were performed using a Bruker DRX-600 Avance spectrometer at 600.13 MHz for 1H, equipped with an xyz gradient unit. Spectra were processed by Bruker XWinNMR software (version 2.5) and MestReNova 6.0 2-5475 software. All samples were prepared in D_2_O and purified by the dialysis procedure.

### 3.6. Attenuated Total Reflection Fourier Transform Infrared Spectroscopy Analysis (ATR-FTIR))

The interactions between citicoline water-soluble compounds and the lipid membranes were evaluated via attenuated total reflection Fourier transform infrared spectroscopy (ATR-FTIR). ATR-FTIR spectra were recorded via a Nicolet IS50 FTIR spectrophotometer (Thermo Nicolet Corp., Madison, WI, USA) equipped with a single reflection germanium ATR crystal (Pike 16154, Pike Technologies, Madison, WI, USA). OMNIC software (OMNIC software system Version 9.8 Thermo Nicolet, Monza, Italy) was used for spectra collection and manipulation. Each spectrum was scanned in an inert atmosphere within the range of 4000–750 cm^−1^ over 64 scans at a resolution of 4 cm^−1^. All sample solutions were placed on the germanium ATR crystal and dried to eliminate water absorption. The ATR-FTIR spectra were performed in triplicate at room temperature.

### 3.7. Time-of-Flight Secondary Ion Mass (ToF-SIMS)

A TRIFT III (Phi Electronics, Chanhassen, MN, USA) mass spectrometer time-of-flight secondary ion mass spectrometer equipped with a 22 keV Au^+^ primary ion beam gun with a beam current of 600pA at an incidence angle of 45° was used to collect a series of mass spectra. The samples were deposited on silica chips and dried under nitrogen flow, then conditioned overnight in a prechamber (vacuum value of 10^−4^ Pa) and then moved to the analyzing chamber (vacuum value to 10^−8^ Pa). The scanning area of secondary ions was 100 µm × 100 µm. Selected peaks were used to calibrate positive ion spectra in the low-mass region: CH_3_^+^ (15.023 *m*/*z*), C_2_H_3_^+^ (27.023 *m*/*z*), C_2_H_5_^+^ (41.039 *m*/*z*).

### 3.8. Release of Citicoline versus Time in the Ophthalmic Simulated Fluid

The ophthalmic simulation fluid was prepared using an oil-in-water emulsion. This emulsion was prepared with castor oil as the oil phase and polysorbate 80 as the emulsifier using the following compositions: castor oil 0.05 %*w*/*v*, polysorbate 80 4.0 %*w*/*v*, glycerol 2.2 %*w*/*v*, sodium acetate 0.05 %*w*/*v*, and chitosan 0.75 %*w*/*v* [33,34].

An amount of 200 μL of liposomes containing citicoline as first placed in Amicon centrifuge tubes (cutoff 3 kDa), then 800 μL of physiological fluid obtained by emulsion oil/water was added to the tube and centrifuged at 5500 rpm for 1 h in a refrigerated centrifuge at 277 K (Thermo Scientific SL 16R, Monza, Italy). An amount of 100 μL was withdrawn from the aqueous phase over time (after 1, 2, 3, 4, 5, and 6 h). Before each collection, the sample was recentrifuged at 5500 rpm for 1 h. These aliquots were used to determine the concentration of citicoline released by UV–visible spectra (at 298.15 K using a Perkin-Elmer Lambda 25 spectrophotometer). Quartz cuvettes with a 10 mm path length were used. A calibration curve was constructed by measuring the absorbance of solutions of known citicoline solution in PBS at 272 nm.

## 4. Conclusions

Many ophthalmic drugs, such as citicoline, are considered to have pharmacological activities against glaucoma, as already demonstrated in various experimental in vitro and in vivo studies. Nevertheless, low aqueous solubility and poor oral bioavailability limit the use of these compounds as effective therapeutic agents. Liposomes represent a category of vesicles that can act as nanocarriers for drug delivery and increase the solubility and stability of drugs, thereby improving their bioavailability and therapeutic potential. Although the entrapment efficiency of citicoline is still relatively low, the improved solubility and optimized efficiency of the encapsulated biomolecules could lead to the development of efficient clinical products.

In the present work, DOPC:Chol liposome preparation containing citicoline were synthesized and characterized. Determination of the average size of liposomes showed a slight increase in the mean size of the diameter of liposomes loaded with citicoline. These data, evaluated together at encapsulation efficiency, indicate that this molecule was successfully integrated into the DOPC:Chol liposomes.

The overall charge of the liposomes showed a slight, statistically nonsignificant change, indicating that the citicoline did not alter the typical bilayer conformation of the liposomes. The NMR, ATR-FTIR, and ToF-SIMS experiments indicated that the citicoline was loaded correctly into the vesicles and that no differences between the two formulations were apparent. Cholesterol concentration affects the release of the drug in the ophthalmic fluid: a high concentration accelerates the release of the drug. The slower release of citicoline in the liposome, with the low amount of cholesterol, allows the drug to act more effectively to protect the optic nerve over time and reduce the administration of eye drops.

## Figures and Tables

**Figure 1 ijms-24-16864-f001:**
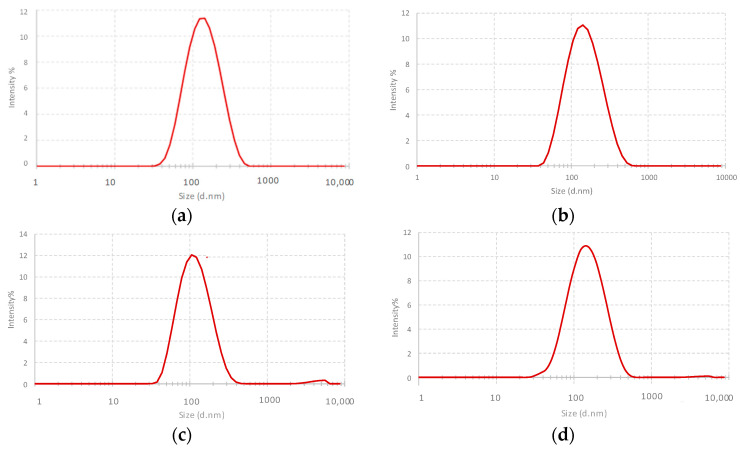
The size distribution by DLS experiments for (**a**) DOPC:Chol (1:1); (**b**) DOPC:Chol, Cit (1:1), 1; (**c**) DOPC:Chol (2:1); (**d**) DOPC:Chol, Cit (2:1), 1 liposomes. The main size distribution is obtained as the average of three replicates (*n* = 3).

**Figure 2 ijms-24-16864-f002:**
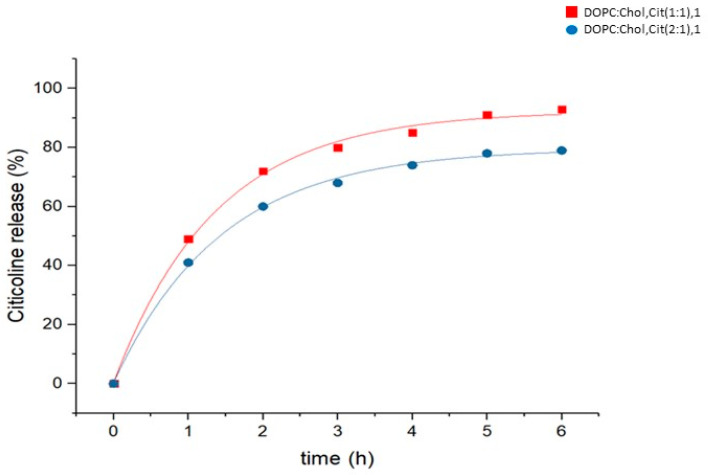
Release of citicoline over 6 h in simulated ophthalmic fluid and first-order fitting data The R^2^ coefficient is 0.998 and 0.999 for DOPC:Chol, Cit (1:1), 1 and DOPC:Chol, Cit (2:1), 1, respectively.

**Figure 3 ijms-24-16864-f003:**
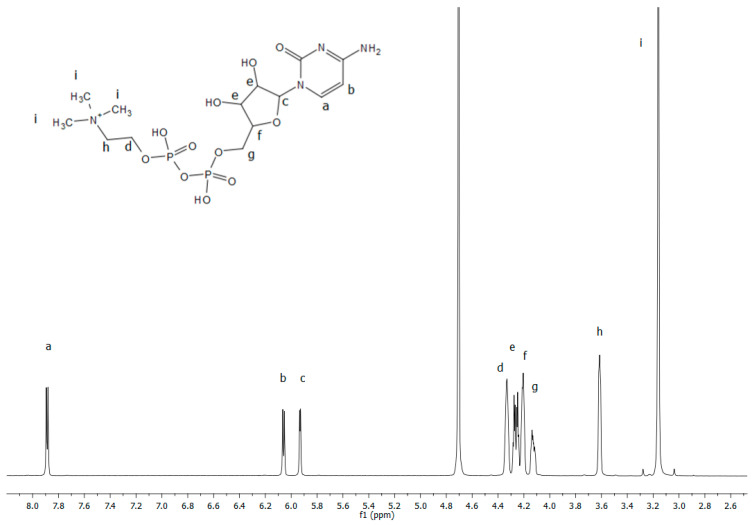
^1^H-NMR spectrum of citicoline 10^−2^ M in D_2_O with proton assignment recorded at 600 MHz and 298.15 K. Chemical structure of citicoline (cytidine-5′-diphosphocholine) with numbering is also reported.

**Figure 4 ijms-24-16864-f004:**
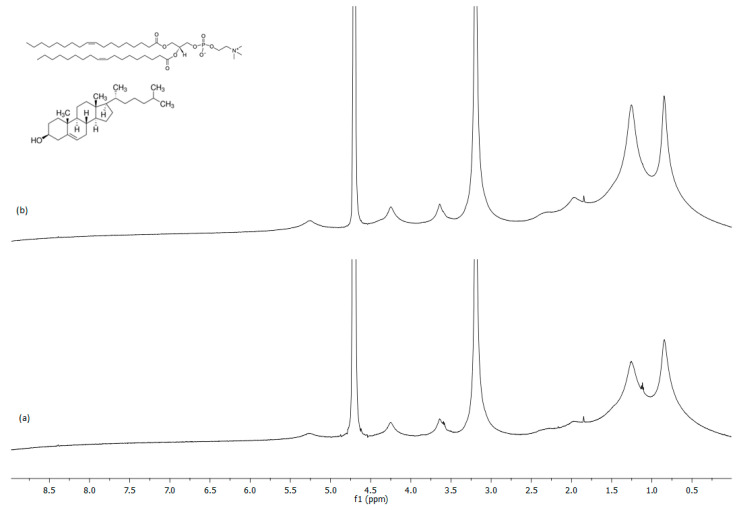
^1^H-NMR spectrum of empty liposomes: (**a**) DOPC:Chol (1:1) and (**b**) DOPC:Chol (2:1) in D_2_O recorded at 600 MHz and 298.15 K. The cholesterol and DOPC chemical structures are reported.

**Figure 5 ijms-24-16864-f005:**
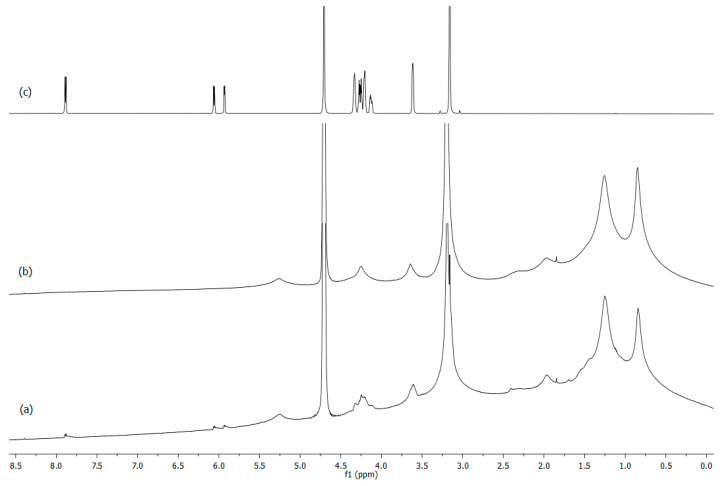
^1^H-NMR spectra of liposomes: (**a**) DOPC: Chol, Cit (2:1), 1; (**b**) DOPC:Chol (2:1); and (**c**) citicoline 10^−2^ M in D_2_O recorded at 600 MHz and 298.15 K.

**Figure 6 ijms-24-16864-f006:**
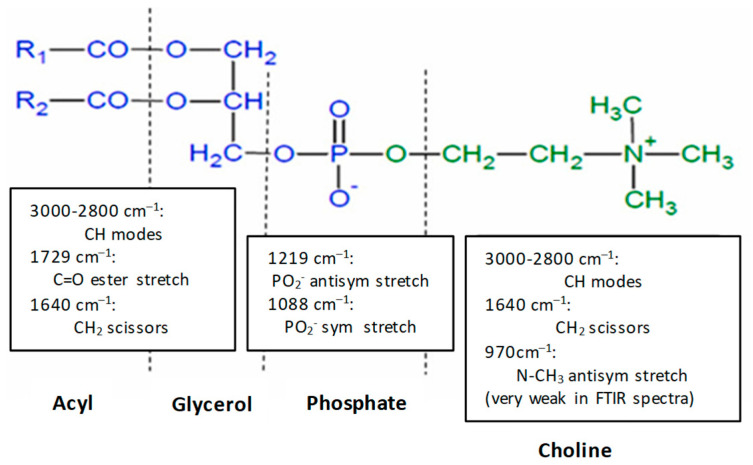
Structure of phosphatidylcholine and the main vibrations of the functional groups identified by FTIR-ATR analysis [31].

**Figure 7 ijms-24-16864-f007:**
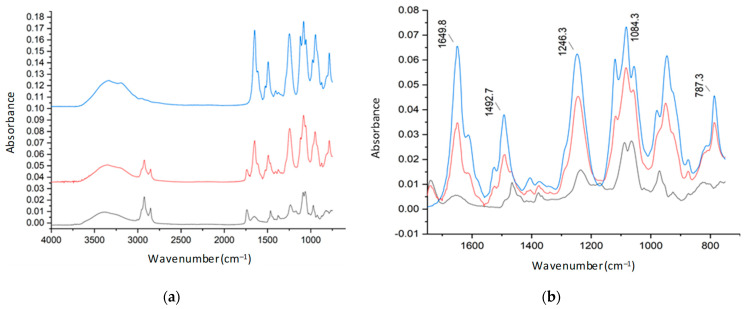
FTIR-ATR spectrum for (**a**) stacked and (**b**) overlapped (1700–750 cm^–1^) DOPC:Chol (1:1) (black); (DOPC: Chol), Cit (1:1), 1 (red); and citicoline (blue).

**Figure 8 ijms-24-16864-f008:**
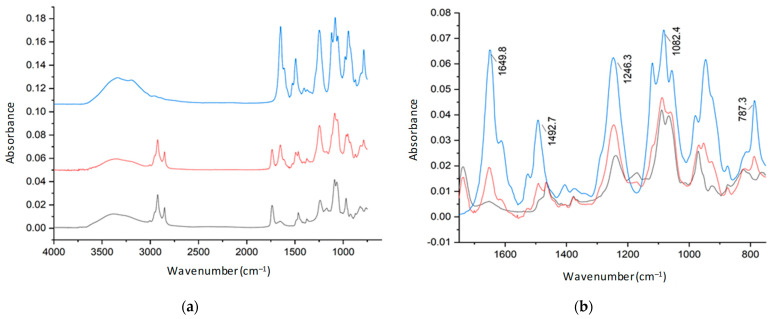
FTIR-ATR spectrum for (**a**) stacked and (**b**) overlapped (1700–750 cm^–1^) DOPC:Chol (2:1) (black); (DOPC: Chol), Cit (2:1), 1 (red); and citicoline (blue).

**Figure 9 ijms-24-16864-f009:**
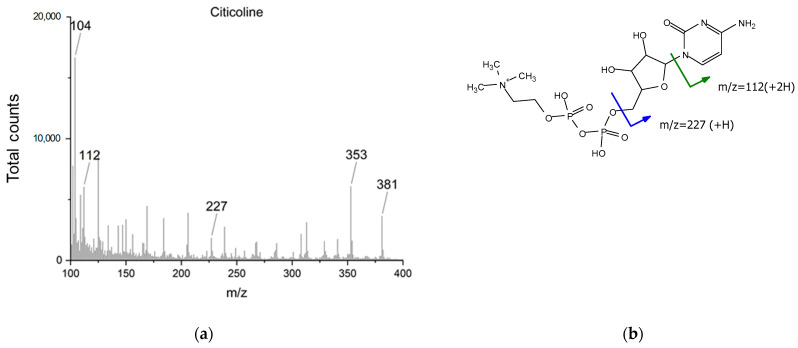
(**a**) TOF-SIMS positive ion spectra of citicoline (*m*/*z* range 100–400); (**b**) structure and characteristic fragments of citicoline.

**Figure 10 ijms-24-16864-f010:**
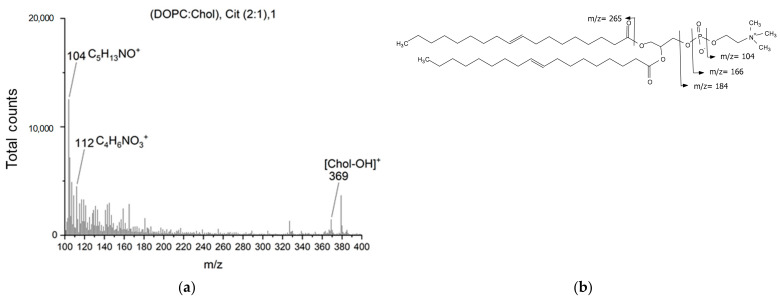
(**a**) TOF-SIMS positive ion spectra of (DOPC:Chol), Cit (2:1), 1 (*m*/*z* range 100–400); (**b**) structure and characteristic fragments of DOPC.

**Figure 11 ijms-24-16864-f011:**
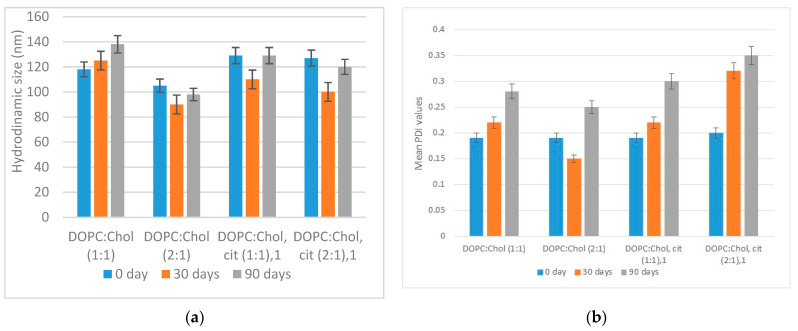
(**a**) Hydrodynamic size of plain and citicoline-loaded liposomes, and (**b**) mean PDI values of plain and citicoline-loaded liposomes over three months at 4 °C (average error 5%).

**Table 1 ijms-24-16864-t001:** Size distribution with polydispersity index (PDI) and ζ-potential for all synthesized liposomes. The values are the average of three measurements (mean ± esd).

Liposome Composition	Molar Ratio	Initial Citicoline Concentration (mol/dm^3^)	Mean Diameter ± SD (nm)	PDI ± SD	ζ-Potential ± SD (mV)
DOPC:Chol	1:1	-	118 ± 4	0.19 ± 0.04	−4.7 ± 0.7
DOPC:Chol	2:1	-	105 ± 3	0.19 ± 0.03	−3.2 ± 0.6
DOPC:Chol, Cit	(1:1), 1	10^−2^	129 ± 4	0.19 ± 0.05	−2.2 ± 0.4
DOPC:Chol, Cit	(2:1), 1	10^−2^	127 ± 2	0.20 ± 0.04	−1.8 ± 0.3

**Table 2 ijms-24-16864-t002:** Encapsulation efficiency (EE%) of liposomes. EE% is calculated using the equation given in the Experimental Section.

Liposome	Molar Ratio	Initial Drug Concentration (mol/dm^3^)	Encapsulation Efficiency ± SD (%)
DOPC:Chol, Cit	(1:1), 1	10^−2^	40 ± 11
DOPC:Chol, Cit	(2:1), 1	10^−2^	47 ± 11

**Table 3 ijms-24-16864-t003:** ^1^H NMR peak assignments of citicoline in D_2_O. The multiplicity is also reported.

Protons	a	b	c	d	e	f	g	h	i
Chemical shift (ppm)	7.89 (d,1H)	6.06 (d,1H)	5.93 (d,1H)	4.31 (d,1H)	4.25 (ddd,2H)	4.20 (S,1H)	4.13 (m,2H)	3.61 (m,2H)	3.16 (S,9H)

**Table 4 ijms-24-16864-t004:** ^1^H NMR peak assignments of DOPC with chemical group indication.

	Chemical Group	Chemical Shift (ppm)
Methylenic protons of DOPC lipid tail	-CH_2_-	0.8
Methyl protons of DOPC lipid tail	CH_3_-	1.2
Methyl group of lipid head	CH_3_-	3.2
Methylenic protons of DOPC lipid head (choline)	-CH_2_-	3.6
Methylenic protons in glycerol part of DOPC head	-CH_2_-, -CH	4.3

**Table 5 ijms-24-16864-t005:** Main bands and assignment observed in FTIR spectra of plain liposomes and citicoline.

Wavenumber (cm^−1^)	Vibration	Sample
3006	Stretching =C-H	Plain liposomes
2920–2850	CH_2_ stretching	
1750	C=O ester stretching	
1652	C=C stretching	
1243	PO_2_^−^ antisymm. stretching, C-O-C stretching	
1091	PO_2_^−^ symm. stretching	
970	C-N^+^-C antisymm. stretching	
3250	-OH stretching	Citicoline
1653	Cytosine C=O stretching	
1652	C=C stretching	
1491	Cytosine C=N	
1243	PO_2_^−^ antisymm. stretching	
1091	PO_2_^−^ symm. stretching	
1117	C-O ether stretching	
970	C-N^+^-C antisymm. stretching	
926	C-N^+^-C symm. stretching	
787	NH_2_ wagging	

## Data Availability

Data are available on request.

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
