# Peer review of "Liposomal Encapsulation of Citicoline for Ocular Drug Delivery"

_ijms, 2023, doi:10.3390/ijms242316864_

Round 1

Reviewer 1 Report

Comments and Suggestions for Authors

The work of C. Bonechi, et. al. "Liposomal encapsulation of citicoline for ocular drug delivery" is interesting, it contains new experimental data and results, which can be useful for various biotechnological projects.

However, the work under review cannot be considered fully completed. The authors need a significant revision of their text, a clearer formulation of the goal and conclusions, and careful editing of description of experiments.

1. In my opinion, the authors pay too much attention to the ophthalmic orientation of their work. This is a purely physical-chemical study of the encapsulation of definite chemical compound, which in principle is applied to ocular therapy. No pharmacological and therapeutic tests were carried out by the authors, and one should be more careful in discussing the potential importance of the systems under study in Conclusions. Moreover, there is uncertainty in this section - why talking about oral bioavailability for ophthalmic drugs?

2. Very unfortunate designation of samples. It needs to be simplified.

3. In experiments on light scattering, there is no necessary information on the details of experiment. How were samples prepared for light scattering experiments? How was the solution cleaned from dust or possible impurities? How many repetitions were performed and any additional details of experiment? The numerical values of 122, 142, 164 nm in Fig. 1 are not explained. These are some particle sizes, but why do they differ from the diameters given in Table. 1? And do not correspond to the trend of liposome size changes shown in the Table 1.

4. NMR experiments are rather vaguely described. In my opinion, the NMR data directly indicate the incorporation of the ligand into liposomes by broadening the citicoline resonance lines in combination with liposomes due to the slow rotation of the latters. Why in the captions to Fig. 3 and 4 authors use the past tense (was, were)?

5. I completely misunderstood the purpose of using the ToF-SIMS method. The authors worked with pure commercial preparations, and it is not clear why to determine their exact chemical structure additionally?

6. There are obvious errors in the text. In Section 4.6 in place of ATR-FTIR there is NMR information. By the way, it is necessary to add information about the details of the experiments in sections 4.5 and 4.6.

Comments on the Quality of English Language

Editing of English language is required.

Reviewer 2 Report

Comments and Suggestions for Authors

Bonechi et al. proposed a study on the design, physicochemical and functional characterization of liposomes loading citicoline. The work was well realized by a methodological point of view and well-described, proposing as a good contribution in the field of lipid formulation for pharmaceutical applications. For this reason, it can be considered for publication after minor revisions.

1) Why did the authors select DOPC as phospholipid component and the used cholesterol contents? 

2) Have the authors an idea by a micro-structural point of the role of the cholesterol content in inducing the different release efficiency?

3) The authors reported a stability of liposome with time, although some changes in size were observed. Did the authors exclude any lipids re-arrangements causing a leakage of the bioactive ingredient?

I suggest a general check of the English style in order to refine the paper style.

Comments on the Quality of English Language

I suggest a general check of the English style in order to refine the paper style.

Reviewer 3 Report

Comments and Suggestions for Authors

I have had the opportunity to review the manuscript titled " Liposomal encapsulation of citicoline for ocular drug delivery" submitted to “IJMS”. I must acknowledge that the problem the authors have tackled is indeed intriguing, and I appreciate the innovative approach they have taken to address it.

While I find the paper to hold great promise, I do have a few concerns and questions that I believe need to be adequately addressed before we proceed with the publication process. While the core idea is commendable, there are certain aspects that warrant careful consideration to ensure the robustness and clarity of the paper. Given these concerns, I would like to recommend that the manuscript undergoes a major revision. I believe that with the necessary improvements, the paper could truly shine and provide valuable insights to the readership.

General Points:

The overall quality of English throughout the manuscript could benefit from improvement.

I recommend revising and elaborating on the captions for all figures and tables, ensuring they provide sufficient detail.

It would be valuable to include discussions that explain the observations in each section, using appropriate paragraph lengths to effectively convey the insights.

Consider adding a bold, normal caption line adjacent to the figure details. This could highlight the figure's key significance or convey its main message. An example of effective caption style can be seen in the paper titled "Protein-avoidant ionic liquid (PAIL)–coated nanoparticles to increase bloodstream circulation and drive biodistribution."

Introduction section:

While the content within the introduction section is indeed impressive, there is room for improvement in terms of the English language to ensure a seamless and coherent flow of the introduction.

Physicochemical characterisation:

The caption of Table 1 is unfortunately divided along the left and below the table. Ideally, commencing the caption prior to the table would provide a smoother presentation. Additionally, the table caption would benefit from greater detail to enhance its contextual understanding.

In the top row of the table, the units of zeta potential should be corrected to "mV" (millivolts) rather than "mv."

Typo error has been identified on line 132, which requires correction.

On line 142, the term "physical-chemistry properties" ought to be amended to "physico-chemical properties."

To enhance the manuscript's flow and readability, it is suggested that repetitions similar to those found on lines 135 and 141 be avoided.

Encapsulation efficiency:

Table 2: Consider enhancing the detail in the caption of Table 2 to provide a more comprehensive understanding of its content.

Line 159: For heightened accuracy, it's advisable to specify the temperature as 298.15 K, unless it was consistently maintained precisely at 298 K.

Line 162: Whenever percentage efficiency is mentioned, include the associated standard deviation for comprehensive reporting. This consistency will contribute to the clarity of the data presented.

Author's Consideration: The author is encouraged to delve into a discussion regarding the potential factors contributing to the disparity in drug loading capacity between the two distinct liposome formulations. Such an analysis can provide valuable insights into the experimental outcomes.

Release Study:

Clarify "Kda" as "kDa" on line 176.

the authors are encouraged to engage in a discussion about release kinetics, emphasizing the potential utility of kinetic equations to comprehend the release mechanism and optimize formulation.

in line 398 The authors are suggested to provide a brief detail how they made the simulated ophthalmic fluid.

In Section 4.8, on Page 13, line 376, authors are recommended to furnish additional details regarding the method employed for release study sample preparation, particularly with regard to various time points.

I have a concern regarding the methodology employed for the release study. Typically, a cellophane membrane is used to study the drug release which prevent the drug carrier's release into the media. An inquiry arises: Did the authors assess the NMR and DLS profiles of the supernatant to ascertain the absence of liposomes containing drug molecules? As UV-vis solely detects the drug and not the liposome carrier, this approach might yield misleading outcomes.

Authors are suggested to the evaluate the adequacy of the citicoline dosage encapsulated within liposomes for eye treatment. It's pivotal to ascertain whether the loaded liposome quantity meets the required dosage for efficient drug delivery at the target site. This evaluation could involve a comparison with commercial citicoline formulations, their dosing, and potential side effects.

the author are suggested to explain how varying cholesterol concentrations impact the release of the drug.

NMR study:

In Figures 3 and 4, it is recommended to represent the NMR X-axis in delta ppm for improved clarity.

At line 204, the term "2D dimensional" could be more appropriately than "bi-dimensional."

In line 209, considering the frequency of NMR at 600MHz, the NMR peaks might be more accurately described as aggregated or unresolved peaks, rather than being labeled as "poor resolution."

With reference to line 158, could the authors elaborate on the rationale behind selecting 272 nm for drug detection?

Could the mechanism of drug encapsulation be expounded upon? It would be insightful to understand the types of interactions favoring drug encapsulation and release. Additionally, the inclusion of a schematic or graphic illustrating drug encapsulation and release, based on rational deductions drawn from all the integrated studies (NMR, FTIR, DLS, Release kinetics), could enhance clarity.

It is suggested to include a discussion section that elaborates on the interpretation of the observations. A paragraph of appropriate length would suffice for this purpose.

ATR-FTIR

Comprehensive analysis is required to elucidate the observations in each section and establish meaningful connections between them.

In Figure 6, ensure that the content is centered within the caption. Additionally, consider enhancing the pixel quality of the written text to improve clarity.

For Figures 7 and 8, consider placing the labels A and B within the figures, preferably at the top-left corner of each respective figure.

Correct a typographical error in line 272.

There is a repetition of the same line in line 294: "that the aggregation/flocculation processes not modified the.

Stability studies:

Why do liposomes remain stable even after three months, despite the presence of a considerable repulsive (either positive or negative) stabilizing surface charge?

To ensure better connectivity and comprehension for the reader, it is advisable to position the "Materials and Methods" section after the "Conclusion" section. This arrangement would facilitate a smoother flow of information and maintain the logical progression of the paper.

I look forward to the authors' response to these concerns and their subsequent revisions. Once these issues are adequately addressed, I believe we will be well-positioned to move forward with the publication process.

Thank you for your attention to this matter.

Comments on the Quality of English Language

-

Round 2

Reviewer 1 Report

Comments and Suggestions for Authors

I agree with correction made on my comments. Manuscript can be accepted in the present form.

Author Response

The authors thank the reviewer for the approval of the manuscript.

Reviewer 3 Report

Comments and Suggestions for Authors

I regret to inform you that minimal to no revisions have been made to the manuscript (except introduction part). Given the esteemed quality and impact of the “IJMS” journal, it is crucial to invest effort in enhancing the manuscript's quality.

Despite the authors' claims of revision, major concerns raised during the review process remain unaddressed. Notably, figure captions have not been revised, and crucial technical discussions elucidating phenomena such as the molecular basis of encapsulation, stability of the formulation, and the mechanism of release kinetics are still conspicuously absent.

An example of this is seen in Line 101, where the statement "Cholesterol affects both the fluidity/permeability and the organization (phase and domain formation) of phospholipid bilayers" lacks clear relevance to the current context.

I strongly urge the authors to seriously consider the previous revision comments and incorporate the necessary changes into the manuscript. Failure to do so may result in the rejection of the paper. It is imperative that any claimed changes are substantiated with meaningful revisions, as a lack thereof is considered unprofessional. I request the authors to refrain from such practices and ensure a diligent and thorough revision process.

Round 3

Reviewer 3 Report

Comments and Suggestions for Authors

I've reviewed both the old and new manuscripts. I found the explanation, especially regarding the encapsulation and release mechanism, less convincing. Utilizing techniques like 2D NMR, such as NOESY experiments, or concentration-dependent 1D NMR experiments, or providing logical explanations based on the interactions of different functional groups in the drug and lipid components, supported by relevant literature, and observations obtained could help in answering the molecular basis of encapsulation and release. The absence of such a crucial element, given the reputation and impact factor of IJMS, makes the study appear preliminary. It's disappointing that the DLS figure hasn't been remade and still exists in a poorly representative form.

Round 4

Reviewer 3 Report

Comments and Suggestions for Authors

I've reviewed the documents, and I am satisfied with the author's explanations. I recommend the publication of this paper.